# Traditional Beliefs, Practices, and Migration: A Risk to Malaria Transmission in Rural Nepal

**DOI:** 10.3390/ijerph192416872

**Published:** 2022-12-15

**Authors:** Kiran Raj Awasthi, Jonine Jancey, Archie C. A. Clements, Rohit Kumar Sah, Madan Prasad Koirala, Binaya Chalise, Justine E. Leavy

**Affiliations:** 1Curtin School of Population Health, Curtin University, GPO Box U1987, Perth, WA 6845, Australia; 2Telethon Kids Institute, Perth, WA 6009, Australia; 3Peninsula Medical School, University of Plymouth, Plymouth PL4 8AA, UK; 4National Malaria Program, Karnali Province Field Office, Nepalgunj 21900, Nepal; 5Graduate School for International Development and Cooperation, Hiroshima University, Higasi-Hiroshima 739-8529, Japan

**Keywords:** socio-cultural, malaria, prevention, control, *Chhaupadi*, migration, transmission

## Abstract

The study aimed to explore sociocultural factors influencing the risk of malaria and practices and beliefs towards malaria prevention, transmission and treatment in a remote village in Khatyad Rural Municipality (KRM) of Nepal. A sequential exploratory mixed methods approach was used. Qualitative data were collected through 25 one-on-one, in-depth interviews followed by a face-to-face household survey (n = 218) among people from a village in KRM believed to have a high risk of malaria. Traditional practices such as *Chhaupadi* requiring the seclusion of women during menstruation and post-partum, transhumance, and reliance on traditional healers for the management of malaria were common practices in the village. The household survey found 98.1% of women faced menstrual exile either inside the house or in a separate hut, with 64.2% not having access to Long-lasting Insecticidal Nets (LLINs). Hardships and economic constraints compelled villagers to migrate seasonally for work to malaria-endemic areas in India, thereby exposing themselves to the risk of malaria. Persistent traditional beliefs and seasonal migration could threaten the elimination goals set by the national malaria program.

## 1. Introduction

Nepal has made significant progress in decreasing the number of malaria cases and is on course to reach the target of elimination set for 2025 [1]. However, sociocultural factors such as embedded cultural and traditional practices, living and sleeping patterns and seasonal migration for work have the potential to increase malaria transmission in rural parts of Nepal. In 2018–2019, Nepal reported 1065 malaria cases, comprising 625 imported cases (transmission outside the country), mainly among seasonal migrants to India, and 440 cases from local transmission [1]. Among the 440 locally acquired cases, 175 were from remote Khatyad Rural Municipality (KRM), an Upper River Valley (URV) area of Mugu district (Karnali Province), in the Western Hills of Nepal

Nepal is divided into three parts: the northern mountains; the middle hills; and the Terai plains in the south. Malaria cases are mostly concentrated in the Terai plains. The subtropical climate in the Terai, dense forests, and travel to and from India contribute to a greater risk of in-country transmission [2]. Malaria cases have recently shifted from the Terai to settlements in the rural areas above 2500 m, such as KRM, that are situated along the valleys of the hills and mountains [2].

Sociocultural factors such as living patterns, communal gatherings of males to eat, drink and smoke late into the evenings, and women caring for cattle in the early morning and late evening, increase the risk of mosquito bites and play a key role in disease transmission [3]. Embedded cultural beliefs and traditions affect decision-making and the type of health care family members received [3,4]. In rural Nepal, extended families usually live in the same household [5], and as Nepal is a patriarchal society, decision-making, particularly regarding health management, is predominantly the responsibility of elderly males [6]. Embedded cultural beliefs and traditions affect decision-making and the type of healthcare received can impede the national malaria elimination program [4,7,8].

In parts of rural Nepal, adolescent girls and women are excluded from the family home during their menstruation and the post-partum period, a practice known as ‘*Chhaupadi*’ [9]. The females are deemed impure during this time and forced to live outside the main house in small huts (*Chaugoths*), cattle sheds, or in less traditional families, in a separate room inside the house [10]. The transient exile is also imposed on mothers and their newborns, whereby they are isolated from family members for 10–14 days postpartum [11]. Nights spent outdoors in *Chaugoths* expose women to the risk of extreme weather conditions, animal attacks, and insect bites [12]. Furthermore, a lack of Long-lasting Insecticidal Nets (LLINs) and access to mosquito repellents, leaves women particularly vulnerable to malaria transmission. The Nepalese government introduced a law in 2017 to stop *Chhaupadi*, which includes up to three months in jail or a fine of 3000 Nepalese rupees (~27 USD) [10,11], however the practice still prevails [13]. Consequently, it is important to explore people’s experiences regarding cultural practices, to be able to design more effective malaria interventions in Nepal.

Population mobility also plays an important role in malaria transmission. The movement of the population in and out of Nepal complicates efforts to contain or mitigate the spread of malaria [14]. Seasonal migration by males in remote areas of Nepal is common. The trend increased during the Maoist revolt in early 2000 due to unemployment in rural areas, with males travelling to the malaria-endemic Terai region or cities in India [4]. The seasonal movement of migrants from high-risk malaria areas can trigger and worsen malaria transmission in local communities [15]. Considering these factors, this study aimed to explore the sociocultural factors influencing the risk of malaria, including sleeping practices and beliefs towards malaria prevention, transmission and treatment in a remote village in KRM of Nepal, with the goal of gaining insight into how best to locally tailor malaria interventions to ensure their effectiveness and sustainability.

## 2. Methods

A sequential exploratory mixed methods approach was used to collect both qualitative and quantitative data from residents of a remote Nepalese village. Briefly, an initial phase of qualitative data collection and analysis was followed by a phase of quantitative data collection and analysis. Priority was given to the qualitative aspect of the study, with the quantitative component used to support the generated qualitative themes [16]. The qualitative component used an interpretative phenomenological approach to understand the lived experiences of the Nepalese people in the remote village [17,18].

The study was approved by the Nepal Health Research Council and Human Research Ethics Committee at Curtin University, Perth, Western Australia (HRE2020-070). The Consolidated Criteria for Reporting Qualitative Research (COREQ) guided the reporting of the study: (1) the research team and reflexivity; (2) study design; and (3) reporting the results [19].

### 2.1. Study Setting

The study site Rigga village lies on the border of Karnali and Sudurpaschim Province, in the URV corridor. The village was selected based on its remoteness, history of outbreaks and being classified as a high risk malaria transmission area [20]. Rigga consists of 466 households and a population of 1447. In 2018 and 2019, Rigga recorded 116 and 173 malaria cases, respectively, accounting for 11% of the total cases in Nepal for both years [1,20]. The village is located a one-hour walk from the nearest road, and it is not accessible during the monsoon season (June–August) due to rain, and in winter (January and February) due to snowfall. The nearest public Health Post (HP) is a half-hour walk from the village. Free malaria testing and treatment is available at the HP.

### 2.2. Sample Size

Recommendations for sample size for non-probabilistic, purposive sampling can range from five to 25 participants, while saturation can occur within the first 12 interviews [21]. Based on this, twenty five participants from the community were purposefully selected for the qualitative arm. For the cross-sectional survey, the sample size (n = 218) was calculated using the sample size calculator assuming significance at 5% and power of 80% [22].

### 2.3. Participant Eligibility and Recruitment

Participants recruited for the one-on-one interviews (for the qualitative component of the study) had a range of ages (15–72 years) and occupations. Household heads (or nominated household decision-makers) were recruited for the household survey (for the quantitative component). One person per household took part in the survey (males and females alternated in every other household).

### 2.4. Development of Measurement Instruments

A semi-structured interview guide was developed based on similar research conducted elsewhere, in addition to the researcher’s prior experience in the field of malaria [20,23]. Public health research has used in-depth interviews as a standard method to explore peoples’ perspectives and experiences [24]. Pilot testing of the semi-structured interview guide was undertaken (n = 3) to ensure the questions were understandable and they informed the study aims. The content was assessed, and questions were modified or excluded based on this process. The final semi-structured interview guide enabled the interviewer to probe and explore areas related to the study.

The household survey was adapted from a study by [25] and piloted (n = 8) in a similar population in a nearby village. For the current study, a subset of data was derived from the self-reported responses of the household surveys. The variables used in this study included: demographics; household structure; sleeping practices and beliefs about malaria prevention and treatment in the village.

### 2.5. Data Collection

The one-on-one interviews were conducted by a trained researcher (KRA) via video conferencing due to COVID-19 travel restrictions. Interviews ranged from 25 to 35 min in length, were conducted in the local Nepali language and audio recorded. Two trained Nepalese enumerators (RKS, MPK) recorded responses to the household survey using a customized Qualtrics application on an android tablet. Informed consent (written and verbal) was obtained prior to any data collection, along with permission to record the interviews. Anonymity of the participants was maintained by using pseudonyms and removing any identifiers.

### 2.6. Data Analysis

Braun and Clarke’s six-stage inductive thematic analysis was used to identify and generate the themes and sub-themes from the data [26]. The recorded interviews were transcribed verbatim in Nepali and then translated into English by the lead author (KRA). All transcripts were read several times thoroughly and coded using N-Vivo software version 12. An open coding process was used to create the codes. The initial coding concentrated on the narrative and the participants’ language. The generated codes were categorised according to similarities and grouped into core themes and subthemes. KRA, JEL and JJ independently identified themes. Consensus was reached on the final themes by three authors (KRA, JEL, JJ). Concurrently, literature on malaria and its determinants was read throughout the data analysis phase to understand and further explain the generated themes. The quantitative data were imported into STATA (StataCorp. 2021. Stata; Release 17). Descriptive statistics were used to summarise the demographic and household characteristics, sleeping practices and beliefs towards malaria treatment of the respondents.

## 3. Results

### 3.1. Qualitative Themes and Participants

The interviews (n = 25), including illustrative quotes, are the focus of the research, and are complemented by the household survey data (n = 218). The qualitative analysis identified two themes: (1) tradition, culture and social taboos; and (2) hardships, necessity and consequences. Appendix A provides an overview of the themes. Interview participants were aged 15 to 72 years. Most (n = 15) resided in an extended family household of three or more generations and had more than five members (n = 20). Appendix A provides an overview of qualitative participants’ socio-demographic characteristics.

Similar numbers of males (n = 108) and females (n = 110) took part in the quantitative survey. More than half of the participants were dependent on agriculture (n = 120), whilst more than one fourth of them were seasonal migrant workers (n = 50). Two-thirds of the participants were literate, however only one third attended schools. Most of the families (n = 185) had a family income of less than 10,000 NRs (~77 USD). More than half of the families were nuclear families (n = 117), however more than half of the families were either large (6–10 members) (n = 98) or very large (<11 members) (n = 16). Often the key decision makers were males in the households (n = 151). Appendix A provides an overview of the socio demographic characteristics of the survey participants and households.

### 3.2. Tradition, Culture and Social Taboo

#### 3.2.1. Traditional Medicine

All participants (n = 25) noted that traditional practices and culture played an important role in shaping their health behaviours. Visits to traditional healers, known as ‘*Dhamis*’ or ‘*Jhakris*’, were an accepted practice for managing malaria. An elderly farmer shared his childhood experience of how a *Dhami* treated him with the ‘*blood of a black sheep*’. The traditional practices for treating malaria extended to consuming other animals (frogs, crabs) and herbs.

“*When there was no health post, the ‘Bhotes’ (Tibetians from the mountains) brought Kuchilo [a type of herb] and told us to eat it. There was one frog called ‘Hasso’ in the rivers; they tied it up and brought it for us to eat. In Terai [plains to the south], they brought crabs, and we were made to eat them*”.(female, teacher)

Some participants (n = 6) cited the common use of traditional herbs or bitter plant leaves from the ‘*Leks*’ (higher lands on hilltops) to drive away the fever due to malaria. However, a shift in treatment choices was observed from the household survey. Majority of the respondents (98.2%) reported that malaria could be cured. Most (84.9%) agreed malaria could not be ‘self-treated’, 92.7% agreed that complications might occur if antimalarial medications were not taken regularly, and 78.9% agreed that antimalarial medication could be bought over the counter at the local pharmacy (Appendix A).

According to the participants (n = 16), malaria has been present in the village for decades:

*“Fever used to come, we used to call it Aula [malaria] or Paljowr. There used to be Bela fever (typhoid) as well”*.(male, photographer)

*“Not a single house was spared. Two to three people from one home had this (malaria), since the time of settlement. If it (malaria) got better in autumn, it would come again in winter”*.(male, ward chair)

A need for approaches to stop traditional practices that may increase malaria was expressed during the interviews. Participants (n = 4) proposed that external disease experts conduct awareness programs on sensitive topics to enlighten the community about malaria transmission and other health risks due to Chhaupadi. They reported that the community would believe external experts compared to local health workers and volunteers.

*“Some people still deny that Chhaupadi increases the risk of malaria in the community. But if people from outside the village, like you, come and tell them, they will listen to you more”*.(male, serviceman)

Several participants (n = 6) suggested that female health workers and members from the mother’s group should take the lead. Others felt (n = 3) it was more important to convince the elders, the priests, traditional healers and teachers of the need for malaria treatment, prevention and control.

*“We have to give education [malaria] programs to the Dhamis and Jhakris (traditional healers) and people who do such things (…). Teachers should also be taught then they will tell it to the children. The children studying in grades 8, 9 and 10 can then go home and tell their mothers and fathers. The elected leaders and community volunteers must be taught these things”*.(male, photographer)

Culturally acceptable activities, such as conducting rallies for advocacy were also proposed for generating interest at the village level *“everyone needs to attend the rally to change the ongoing practice and adopt new practices”*(female, housewife). A student proposed culturally appropriate and interest generating strategies for example:
*“There are also things like discussion and street drama. Also, ‘Deuda’ (a communal dance with songs) and programs for malaria”*.(female, student)

#### 3.2.2. Transhumance

All participants spoke about transhumance, (i.e., taking cattle and livestock up to the sheds in the ‘*Leks’* to avoid being bitten by mosquitoes in lowlands, known as *‘Aul’*). Several participants (n = 3) cited the need for cow manure to cultivate crops in the *Leks* as the reason for the transitional shifting of the animals from July to August.

*“We would plant millet, maize, and wheat, so it would be far to fetch the manure (fertilisers), so for that, we would be going there (to the Leks) since our forefathers”*.(female farmer)

#### 3.2.3. Chhaupadi

A common practice in the area was ‘*Chau*’ or ‘*Chhaupadi*’. All participants accepted that ‘*Chau’* was still practiced in some way in the community. The household survey also revealed that menstrual exile persists, with almost all respondents (98.1%) reporting that women were separated from the main household during menstruation (Appendix A). Some interview participants did stress that the practice had changed over time (n = 9). Participants highlighted that women were often afforded a clean separate room within the house.

*“During menstruation, there is a room near the kitchen; we sleep there (separate room) and do not touch the kitchen for five days. We are provided with food from far away. After five days, we wash our clothes and enter the kitchen”*.(female, housewife)

Some of the participants (n = 8) stressed that currently, women had access to a bed and LLINs; “*We (females) sleep in a separate room with a bed and use mosquito nets, but the room has no electricity”* (female, student). However, almost two-thirds (64.2%) of the surveyed households reported that they did not provide LLINs to women during their menstrual exile (Appendix A). The current situation as described contrasted earlier times when women slept outdoors in sheds (*Chaugoths*) far away from the house (Appendix A). An educated older female participant shared her past *Chau* experience to the present:
*“I started menstruating when I was 13 years old. In my paternal house, they didn’t allow me to touch the pillars (columns) of the house (…) they made us stay in a cowshed where there were no doors or windows. I was also made to stay in a cave far away from the house… many times. We had to stay the whole night alone. My second mother-in-law [aunt-in-law] died in a cowshed (Chaugoth). She had a fever for three days. Now different organisations and the Nepal Government have raised awareness to end the Chhaupadi tradition. They (family) allow us to touch parts of the house. Some women have a separate room at home and sleep there”*.(female, teacher)

The participants explained that the exile would be different for menstruation and post-partum including the duration.


*“It’s 10 days for them (postpartum mothers) and five days for menstruating women. In the sixth day we take a bath and can carry out puja (worship) in the room where the gods are kept” shared a female community health volunteer.*


Most of the participants (n = 19) agreed that during the menstrual and post-partum phase, women were not allowed to visit temples, touch males in the household and were barred from consuming products derived from cow’s milk (a sacred animal in the Hindu culture). In recent years, some changes have been observed.

“Nowadays buffalo milk has been permitted to menstruating women, whilst cow’s milk was allowed to the secluded post-natal mothers” (female, housewife). An elderly female shared her experience “Post-natal mothers are accompanied by a family member while staying separately. If the husband is there, he goes to stay with her. If the husband is not there, the mother-in-law, sister-in-law or anyone else stays with her. She is not left alone.” (female, housewife)

Reasons for the seclusion were to appease the Gods; “some people believe that it is bad if they [menstruating women] come near the house where gods and goddess are kept because we feel the Gods will be angry and harm us.” (male, farmer). Participants indicated that the tradition has waned in recent years. A young woman cast doubt on the claims about Chau; “I don’t even tell anyone about my menstruation, and I eat what I cook myself, and I sleep in my bedroom.” (female, hotelier).

### 3.3. Hardships, Necessities, and Consequences

#### 3.3.1. Living Conditions

The difficult living conditions in rural Nepal were discussed, including living in houses built of stone and mud. These two-storied houses have cattle sheds on the ground floor, and on the second floor, where the family live, there is usually two to four rooms, with no demarcation between the bedrooms and kitchen. The participants explained that the houses were small with limited bedrooms. All participants indicated that they had two houses, one in the *Aul* (lowlands along the river valleys) and another in the *Leks* (the highlands in the mountains). Often the entire family would sleep together due to a lack of space. One participant commented; *“Almost everyone has to sleep together (laughs) because there are few rooms.”* (male, contracted serviceman).

#### 3.3.2. Work

Most of the participants (n = 18) were farmers who relied on agriculture for their livelihood. Whilst they worked throughout the year from dawn until dusk, their income was insufficient. To supplement their income, migration to India to work was common;

“The youngest son works in Bombay and sends money. The food grown in our farm is not enough for the whole year, so we need to buy rice” (female, housewife/mother).

The same sentiment was shared by an elderly mother who highlighted the need to migrate for work to make ends meet for the family *“I took out a loan and sold my land. Later we went to India and earned money*” (female, housewife/farmer). Participants who did not have land depended on others for their food

“We are poor. We go and work for others, and they pay us with grains” (female, housewife/farmer).

Participants in the village noted that seasonal movement had existed for many decades. Financial hardship and lack of employment opportunities in the village caused young males to migrate to the urban areas in the Terai or across the border to India for work.

“They go to India in Kartik and Mangsir (November/December) and return on Falgun and Chaitra (March/April)” (female, farmer/housewife).

Participants were unaware of the health issues migrant workers might create, although a few (n = 3) did indicate they could bring back malaria. India was the most preferred destination for the seasonal migrants;

“People often go to India. Recently, 50–60 people from our village have gone (to India)” (male, priest).

#### 3.3.3. Healthcare Access

The remoteness of the village and poor access to tertiary healthcare facilities at times forced families to secure loans at high-interest rates to cover healthcare costs, exacerbating their economic hardships. Almost four out of ten (38.5%) households surveyed reported cost as the deciding factor when seeking care for malaria (Appendix A). Many participants (n = 18) stated that the cost of advanced medical care, including travel and lodging, ranged between Nepalese Rupees (NRs) 25,000 (~USD 200) and NRs 370,000 (~USD 3000). Major illness, including severe malaria, in multiple members of the family, would plunge the family into financial turmoil. For the participants, the value of human life outweighed any hardships the family had to endure.

“We sold the farm and two cows. We also used the middle son’s earnings. We can’t let him die; we have to save him anyway” (female, housewife/farmer).

“We had to sell our shop, and all the goods from the shop. Yet, our grandson recovered, which is bigger than anything” (elderly male, farmer).

## 4. Discussion

This study aimed to explore sociocultural practices and beliefs related to the risk of malaria prevention, transmission and treatment in a remote rural area of Nepal. The results identified the existence of traditional practices and beliefs known to be associated with an increased risk of malaria transmission. These included *Chhaupadi*, transhumance and seasonal migration for work to areas with high levels of malaria. Drivers of these practices were tradition, lack of work and financial necessity.

It is known that sleeping indoors and using LLINs act as barriers to mosquito bites [4]. Yet, the continued practice of *Chhaupadi* increases the risk of malaria, as it compels women and children to sleep outdoors during menstruation and the post-partum period, with minimal protection from malaria transmission. Recent Indian studies have reported that the odds of being infected with malaria among those sleeping outdoors was 70% higher than those sleeping indoors under LLINs [27] and that *Chhaupadi* practices are more prevalent in extended families, due to pressure exerted by elders [10].

Currently, women and children comprise more than three-quarters (77.5%) of malaria cases in KRM [1]. Breaking down long-standing beliefs and practices that increase risk of malaria is challenging, however, research findings have reported a positive shift in recent years with respect to *Chhaupadi* [10,27]. In our study, some interviewees reported a change in this behaviour, indicating that menstrual and post-partum exile among younger women in the community was no longer being practiced. By contrast, almost all (98%) participants in the household survey indicated this practice is continuing. These conflicting results could be attributed to people misreporting due to fear of the existing *Chhaupadi* laws, whereby family members enforcing *Chhaupadi* face fines and up to three months in prison [10,11]. In Nepal, the NMEP distributes free LLINs in Nepal to all high risk households at a ratio of one net for every two person [1]. However, until the law against *Chhaupadi* is enforced and the traditional practices are eliminated, the NMEP must ensure that LLINs are available for all women and children in each household to protect them during periods of seclusion.

This study highlighted the practice of transhumance of livestock for two months every year (July–August), a time period that coincides with the higher incidence of malaria cases in the study area [1]. Malaria vectors can be of two types, anthropophilic (biting humans) and zoophilic (biting animals) [28]. However, in the absence of animals, mosquitoes bite humans, as they need to feed on blood to lay their eggs [28,29]. Malaria vectors (*Anopheles culicifaces* and *Anopheles fluviatilis*) in Nepal, are mainly zoophilic and endophagous (biting indoors), but they have started showing anthropophilic and exophagous (biting outdoors) parasitic behavior [15,29]. Regular vector bionomic monitoring in malaria-endemic areas is needed to understand the biting behaviour of the mosquitoes.

Malaria in Nepal is diagnosed via Rapid Diagnostic Tests, all positive cases reconfirmed by microscopy and treated by antimalarial drugs as per the national malaria treatment protocol [1]. Both the testing and treatment for malaria is free across all public health facilities. However, our results showed that malaria had been in the study site for decades, with participants reporting that they resorted to traditional management through the *Dhamis* and *Jhakris*. The traditional management of malaria is passed on from one generation to the next [30]. Traditional treatments using plant extracts or herbs are not uncommon in rural communities across Asia and Africa due to limited health care services and disbelief in allopathic (Western style) medicines [31,32]. The medicinal properties of plants used by traditional healers should not be underestimated, as plant-based medications used for malaria treatment containing traces of artemisinin and quinine derivatives (constituents of allopathic antimalarial drugs) have been identified [30]. This may partly explain the symptomatic relief from malaria while using traditional herbs. However, *Plasmodium vivax,* the dominant variety of malaria in Nepal, remains dormant in the host’s liver and can reoccur if not treated with a complete two-week course of Primaquine [33,34,35]. Our study results indicated positive beliefs around diagnosis and treatment of malaria from a health facility and the need to complete treatment by taking the full course of malaria medication as prescribed. However, continued awareness raising and access to medication for malaria prevention and control are vital, and need to be ongoing.

Malaria is considered a disease of the poor and is associated with crowded living conditions [8,36]. Participants in our study lived in small two-storey mud houses (Appendix A), and more than 80% of the families were large (>5 members). Higher mosquito densities have been reported in rural houses with mud walls, open eaves, unscreened windows, grass rooves, and near poultry or livestock pens [37,38]. Insufficient rooms, with members sleeping together, make it difficult to use LLINs properly. Furthermore, limited indoor space, and warm weather conditions support outdoor sleeping habits, further increasing the risk of malaria [4,39]. Conversely, changes in living standards and the modernization of housing, with features such as nets in windows, play a role in controlling malaria [37,38]. It is important to make communities aware of the importance of safe sleeping practices, such as always sleeping under LLINs and the role of screened windows.

Economic hardships and limited work opportunities in rural areas lead to seasonal migration, a major factor associated with malaria transmission in Nepal [15]. Annually an estimated 700,000 thousand migrant workers cross over to India from the four western check posts [15]. Furthermore, several studies across the Asia Pacific region support the theory of increased risk of malaria among migrant workers [40,41,42]. Nepalese migrant workers commonly reported working at night as security guards or labourers in malaria-endemic areas of India, increasing their risk of mosquito bites [15]. Accordingly, occupation and the subsequent duration of exposure to vectors in the workplace are closely associated with malaria transmission among migrant workers [40,41]. Going forward, a multi-component strategy should be adopted to minimize malaria transmission risk among migrant workers. Potential strategies should include: prophylactic malaria treatment with mefloquine prior to travelling to malaria endemic countries for work [43]; provision of LLINs to migrant workers to take to their workplaces in India to use in their temporary dwellings [44]; community-based testing for migrant returnees, for early detection and prompt treatment [15]; and raising awareness of the importance of mosquito repellents, LLINs and the wearing of protective clothing during late working hours [40,41,42].

### Strengths and Limitations

To our knowledge, this is the first study to explore the beliefs and practices, including *Chhaupadi* and migration, associated with malaria transmission in rural Nepal. Due to COVID-19 travel restrictions, the interviews were conducted virtually; a limitation at times due the poor internet services in the areas that compromised the flow of the qualitative interviews. Nonetheless, the interviews had a similar feeling as that of conducting a face-to-face interview in terms of the communication and conversation. Both the interviews and the survey tools were pre-tested and the interviews were conducted in the local language, which allowed the participants to express their views openly and fully.

## 5. Conclusions

The study provides insights into practices and beliefs associated with the increased risk of malaria transmission in rural Nepal. Traditional practices such as *Chhaupadi*, transhumance, traditional healers, and poor housing design have the potential to stall the progress made to eliminate malaria in Nepal. Remoteness, financial hardship and lack of employment at the local level are key drivers for the continued seasonal migration of people to malaria-endemic areas of India. Improved housing, access to LLINs, and culturally appropriate strategies that address sociocultural factors could elicit positive behaviour change among rural Nepalese people, thereby improving malaria prevention and treatment in the community and reduce malaria transmission.

## Data Availability

The datasets generated for this study can be found in the Qualitative data repository at https://doi.org/10.5064/F6WMOBYB.

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
