# Peer review of "Traditional Beliefs, Practices, and Migration: A Risk to Malaria Transmission in Rural Nepal"

_ijerph, 2022, doi:10.3390/ijerph192416872_

Round 1

Reviewer 1 Report

This study by Awasthi and colleagues provides interesting research about traditional beliefs, practices, and migration in rural Nepal. The study presents important results for the country's health services in relation to malaria control. I have a couple of fairly major issues to resolve before I can recommend publication, but I think there's a good paper here.

Introduction

- Lines 43, 76: Once the acronym has been described (line 36), there is no need for continued explanation.

- Lines 63 and 328: Up to three or six months? The information needs to be consistent.

Methods

- Line 92: Authors should describe the study site in more detail: total village population, average annual malaria cases, types of treatments available, etc.

- Line 101: It is important to define the study inclusion and exclusion criteria. Was it all the villagers? Was 1 person selected per household? How was the sample size calculated? All this information must be properly described.

- Lines 109 and 113: Interestingly, the authors described that pilot tests were carried out to adjust the interviews. However, it is necessary to inform how they arrived at this n=3 and n=8; What were these values based on?

Results

- Lines 143-144: How did the authors arrive at this number? It is very important to describe in the methods section how the selection of participants was carried out.

- Line 228: It is not necessary to repeat the meaning of the acronym.

- Line 231: Table x?

Discussion

- Lines 307-308: The authors reinforce that “the aim of the study was to explore sociocultural practices and beliefs related to the risk of malaria prevention, transmission and treatment in a remote rural area of Nepal.” However, there is no information on malaria prevention and transmission. Would the authors have more information, and could they write more about these two areas?

- Line 318: LLNs or LLINs

- Lines 351-352: P. vivax is not a mosquito! Please correct this information.

- Line 354: The authors did not comment on how the diagnosis is made. Could the authors clarify?

- Line 360: This figure is mistakenly identified as 2 in the supplementary material.

- Line 376: Authors must standardize according to the journal's rules: parentheses or square brackets.

General comments

- Please, the authors should carefully read the entire text and check that the sentences (referring to the transcripts of the participants' speeches) are correctly in italics.

- It was not clear whether malaria treatment and LLINs are distributed free of charge or whether it is necessary to pay for them. The authors could add this information so that the reader can have a better idea of the context of this population.

Author Response

Response to reviewer’s comments

Manuscript title: Traditional beliefs, practices and migration: A risk to malaria transmission in rural Nepal

Dear reviewer,

Thank you for your comments and suggestions which have been addressed and colour coded in the manuscripts for your kind perusal. We would also like to acknowledge the effort and time that you have provided to go through our manuscript.

Introduction

Comment 1

- Lines 43, 76: Once the acronym has been described (line 36), there is no need for continued explanation.

Response 1

Thank you for the comment. This has now been corrected. Please refer to page 2, line 42, 76.

Malaria cases have recently shifted from the Terai to settlements in the rural areas above 2,500 meters, such as KRM, that are situated along the valleys of the hills and mountains

Comment 2

- Lines 63 and 328: Up to three or six months? The information needs to be consistent.

Response 2

Thank you for the comment. This has been corrected. Please refer to page 8 line 349

These conflicting results could be attributed to people misreporting due to fear of the existing Chhaupadi laws, whereby family members enforcing Chhaupadi face fines and up to three months in prison

Methods

Comment 3

- Line 92: Authors should describe the study site in more detail: total village population, average annual malaria cases, types of treatments available, etc.

Response 3

Thank you for the comment. The information has been added. Please refer to page 3 lines 94-101.

Rigga consists of 466 households and has a population of 1,447. In 2018 and 2019, Rigga recorded 116 and 173 malaria cases respectively, accounting for 11% of the total malaria cases in Nepal for both years. The village is located a one-hour walk from the nearest road, and it is not accessible during the monsoon season (June- August) due to rain, and in winter (January and February) due to snowfall. The nearest public Health Post (HP) is a half-hour walk from the village. Free malaria testing and treatment is available at the HP.”

Comment 4

- Line 101: It is important to define the study inclusion and exclusion criteria. Was it all the villagers? Was 1 person selected per household? How was the sample size calculated? All this information must be properly described.

Response 4

Thank you for the comments. The sample size calculation for both arms has been added as suggested. Please refer to page 3 line 103-110, and 116.

2.2. Sample size

“Recommendations for sample size for non-probabilistic, purposive sampling can range from five to 25 participants, while saturation can occur within the first 12 interviews [21]. Based on this, twenty five participants from the community were purposefully selected for the qualitative arm. For the cross-sectional survey the sample size (n=218) was calculated using the sample size calculator assuming significance at 5% and power of 80% [22].

“One person per household took part in the survey (males and females alternated in every other household).”

Comments 5

- Lines 109 and 113: Interestingly, the authors described that pilot tests were carried out to adjust the interviews. However, it is necessary to inform how they arrived at this n=3 and n=8; What were these values based on?

Response 5

Thank you for the comments. The pilot testing for the interviews were done on 10 percent of the total sample. However, for the quantitative survey, four percent of the total sample was piloted as the questions were sourced from a previous study [20] and the pilot aimed to assess the clarity and coherency of the questions.   

Results

Comments 6

- Lines 143-144: How did the authors arrive at this number? It is very important to describe in the methods section how the selection of participants was carried out.

Response 6

Thank you for the comments. This has been added. Please refer to response 4 above

Comments 7

- Line 228: It is not necessary to repeat the meaning of the acronym.

- Line 231: Table x?

Response 7

Thank you for your comments. This has been revised. Please refer to page 6 line 242 and 245

“Some of the participants (n=8) stressed that currently, women had access to a bed and LLINs;

However, almost two-thirds (64.2%) of the surveyed households reported that they did not provide LLINs to women during their menstrual exile (Table 2, supplementary file).”

Discussion

Comments 8

- Lines 307-308: The authors reinforce that “the aim of the study was to explore sociocultural practices and beliefs related to the risk of malaria prevention, transmission and treatment in a remote rural area of Nepal.” However, there is no information on malaria prevention and transmission. Would the authors have more information, and could they write more about these two areas?

Response 8

Thank you for your comment. We have discussed about prevention and transmission of malaria in the results and discussion. In the results and discussion we talk about

  1. Chaupadi- where 64.2% of the surveyed households reported that did not provide LLINs (important preventive measure) to women during their menstrual exile (please refer to page 6 line 243), and in the discussion (page 8 line 334-339)
  2. The role of transhumance on increased risk of malaria transmission is also highlighted (please refer to page 5 line 222-229, and page 8 line 355-)
  3. Similarly, the role of migration in malaria transmission has been discussed (please refer to page 9 line 398)

Comments 9

- Line 318: LLNs or LLINs

Response 9

This typographical error has now been corrected. Please refer to page 8 line 339.

Recent Indian studies have reported that the odds of being infected with malaria among those sleeping outdoors was 70% higher than those sleeping indoors under LLINs

Comments 10

- Lines 351-352: P. vivax is not a mosquito! Please correct this information.

Response 10

Thank you for the comment. This has now been corrected. Please refer to page 8 line 375

However, Plasmodium vivax, the dominant variety of malaria in Nepal, remains dormant in the host's liver and can reoccur if not treated with a complete two-week course of Primaquine

Comment 11

- Line 354: The authors did not comment on how the diagnosis is made. Could the authors clarify?

Response 11

Thank you for the comment. A line has been added for clarity. Please refer to Page 8 line 365-367.

“Malaria in Nepal is diagnosed via Rapid Diagnostic Tests with all positive cases reconfirmed by microscopy and treated by antimalarial drugs as per the national malaria treatment protocol [1].”

Comments 11

- Line 360: This figure is mistakenly identified as 2 in the supplementary material.

Response 11

Thank you for the comment. This has been corrected in the supplementary file.

Comments 12

- Line 376: Authors must standardize according to the journal's rules: parentheses or square brackets.

Response 12

Thank you for the comment. This has been corrected. Please refer to page 9 line 400.

“Nepalese migrant workers commonly reported working at night as security guards or labourers in malaria-endemic areas of India, increasing their risk of mosquito bites [15].”

General comments

Comment 13

- Please, the authors should carefully read the entire text and check that the sentences (referring to the transcripts of the participants' speeches) are correctly in italics.

Response 13

Thank you for the comment. This has been corrected throughout the paper.

Comment 14

It was not clear whether malaria treatment and LLINs are distributed free of charge or whether it is necessary to pay for them. The authors could add this information so that the reader can have a better idea of the context of this population.

Response 14

Thank you for the comment. This information has been now added. Please refer to page 8 line 350-351 and 367-368.

Both the testing and treatment for malaria is free across all public health facilities.”

“In Nepal, the NMEP distributes free LLINs to all high risk households at a ratio of one net for every two person [1].”

Reviewer 2 Report

1- Line 155 . I suggest the family income to be mentioned in an international currency like dollar as well. 

2- Give a clear definition for some local words such as chhaupadi, please. 

Author Response

Manuscript title: Traditional beliefs, practices and migration: A risk to malaria transmission in rural Nepal

Dear reviewer,

Thank you for your comments and suggestions which have been addressed and colour coded in the manuscripts for your kind perusal. We would also like to acknowledge the effort and time that you have provided to go through our manuscript.

Comment 1

Line 155. I suggest the family income to be mentioned in an international currency like dollar as well. 

Response 1

Thank you for the comment. The value has been provided in USD. Please refer to page 4 line 169.

Two-thirds of the participants were literate, however only one-third attended schools. Most of the families (n=185) had a family income of less than 10,000 NRs (~77 USD).”

Comment 2

2- Give a clear definition for some local words such as Chhaupadi, please. 

Response 2

Thank you for the comment. The definition of Chhaupadi has been provided in the manuscript background. Please refer to page 2 lines 53-55

 Write the definition here

In parts of rural Nepal, adolescent girls and women are excluded from the family home during their menstruation and the post-partum period, a practice known as 'Chhaupadi' [9]”

Round 2

Reviewer 1 Report

I believe that the author made all the necessary changes and corrections to improve the manuscript. I have nothing else to consider.